# LEARNING PERSONALIZED STORY EVALUATION

## ABSTRACT

While large language models (LLMs) have shown impressive results for more objective tasks such as QA and retrieval, it remains nontrivial to evaluate their performance on open-ended text generation for reasons including (1) data contamination; (2) multi-dimensional evaluation criteria; and (3) subjectiveness stemming from reviewers' personal preferences. To address such issues, we propose to model personalization in an uncontaminated open-ended generation assessment. We create two new datasets Per-MPST and Per-DOC for personalized story evaluation, by re-purposing existing datasets with proper anonymization and new personalized labels. We further develop a personalized story evaluation model **PERSE** to infer reviewer preferences and provide a personalized evaluation. Specifically, given a few exemplary reviews from a particular reviewer, **PERSE** predicts either a detailed review or fine-grained comparison in several aspects (such as interestingness and surprise) for that reviewer on a new text input. Experimental results show that **PERSE** outperforms GPT-4 by 15.8% on Kendall correlation of story ratings, and by 13.7% on pairwise preference prediction accuracy. Both datasets and code will be released.

## 1 INTRODUCTION

Large language models (LLMs) have recently shown impressive performance in many tasks (Ouyang et al., 2022; Bai et al., 2022; Touvron et al., 2023), leading to increased interest in benchmarks for determining current models' exact capabilities and limits (Hendrycks et al., 2020; Suzgun et al., 2022; Liang et al., 2022). However, LLMs' capabilities in open-ended text generation are still insufficiently examined due to a lack of reliable evaluation metrics.

Evaluating open-ended text generation, such as long-form question answering and story generation, is challenging due to the one-to-many issue (Liu et al., 2016) and the complexity of long-range coherence and relevance (Yang et al., 2022; 2023). Traditional automatic metrics such as ROUGE (Lin, 2004) have shown poor correlation with human judgment (Krishna et al., 2021; Guan et al., 2021). Meanwhile, recent metrics propose to directly use LLMs as evaluators (Fu et al., 2023; Liu et al., 2023). However, LLM evaluation models have been shown to exhibit biases related to position or length (Zheng et al., 2023; Bai et al., 2023). Besides, the contamination problem may affect the evaluation performance (Chang et al., 2023).

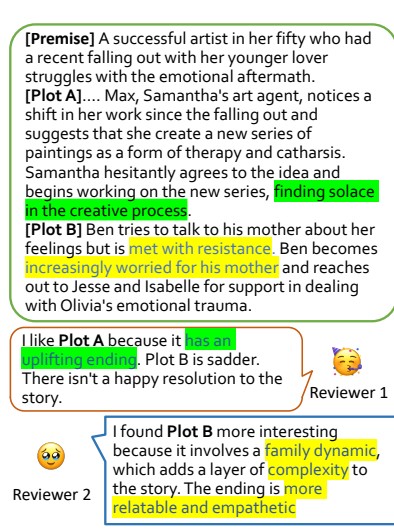

Figure 1: Different reviewers have diverse preferences when evaluating plots. The review and the corresponding plot element are highlighted in the same color.

Human evaluation is also widely used in open-ended text generation. However, it may be time-consuming and expensive, especially for larger-scale evaluation. Moreover, crowdsourced human evaluation can suffer from poor inter-annotator agreement, especially on subjective aspects (Karpinska et al., 2021; Xu et al., 2023a), where disagreement may stem from diverse human preferences. This personalization issue in text generation has recently attracted increasing attention (Flek, 2020; Dudy et al., 2021), but evaluation methods for addressing it are still under-explored.

In this paper, we explore personalized evaluation for long-form story generation, where the assessment is heavily influenced by reviewers' personal preferences. For example, Figure 1 illustrates two reviewers' opinions when comparing two plots derived from the same premise. Reviewer 1 prefers Plot A for its uplifting ending while Reviewer 2 favors Plot B because of the plot complexity and empathetic ending. To model such diverse preferences in story evaluation, it is important to have a standardized personalization evaluation dataset, and learn to evaluate stories from a particular individual's perspective from limited preference data.

**Personlization Evaluation Dataset**. Few story evaluation datasets have personal labels due to the difficulty of collecting personal information. Besides, most existing story datasets have been exposed to LLMs. This leads to a contamination issue for LLM-based evaluation models (Chang et al., 2023). To create less biased story datasets for personalization evaluation, we re-purpose two datasets for personalized story evaluation: Per-MPST from the movie plot-review dataset MPST v2 (Kar et al., 2018) and Per-DOC from the human evaluation results of Yang et al. (2023). We use reviews from the same reviewer as the implicit evaluation preference and elaborate the dataset construction to reduce the memorization of existing stories. Meanwhile, we also use human evaluation results (with annotator identification) of novel stories generated by story generators.

**Reviewer Preference Modeling**. We propose an LLM-based **PER**sonalized **S**tory **E**valuation model (**PERSE**) to capture a particular reviewer's preference given limited personalized data. **PERSE** takes several annotated reviews by the same reviewer as the implicit preference, and uses them to infer a personalized review in the form of a real-valued score plus a text rationale. **PERSE** can also conduct a personalized comparative evaluation of two plots on various fine-grained aspects, such as interestingness and surprise. **PERSE** is trained to infer the given reviewer's preference by instruction-tuning LLaMA-2 (Touvron et al., 2023), and significantly outperforms GPT4 (OpenAI, 2023): **PERSE** achieves a 15.8% higher Kendall correlation in individual story evaluation and a 13.7% higher accuracy in comparative evaluation. Our contributions can be summarized as below:

- We repurpose two story datasets Per-MPST and Per-DOC for personalized story evaluation. By elaborating the data construction, we alleviate the evaluation bias caused by contamination problem in LLM-based evaluation.
- We develop a personalized story evaluation model **PERSE** to predict personalized reviews for new reviewers. It infers a reviewer's preferences from a limited number of prior reviews by the same reviewer, and provides a detailed personalized review for a new story, such as a textual explanation or fine-grained comparison on various aspects. Experimental results show it significantly outperforms GPT-4 on unseen users under both evaluation settings.
- We make several observations on personalization modeling in LLMs. Even given the personal preference, the direct usage of LLMs (such as LLaMA-2 or GPT-4) will still get a generic and less-critical review. Besides, when the context is long, it is difficult for them to benefit from additional personalized examples. However, with instruction-tuning on several thousands of data, these LLMs can effectively gain the capability to align with a new reviewer and become more robust and powerful with more personalized examples.

## 2 RELATED WORK

**Automatic Story Evaluation** Many automatic metrics have been proposed for evaluating language generations. They can be briefly divided into reference-based and reference-free metrics. Reference-based metrics evaluate the similarity between the reference and the model output based on lexical overlap (Papineni et al., 2002; Lin, 2004) or embedding distance (Zhang et al., 2019; Zhao et al., 2019). However, these reference-based metrics have shown poor correlation with human evaluation in open-ended generation due to the one-to-many issue (Liu et al., 2016). Meanwhile, reference-free metrics directly measure the quality of the model output without any reference. Usually, they are trained to distinguish good and bad generations from an overall perspective (Guan & Huang, 2020; Ghazarian et al., 2021) or along multiple axes (Chen et al., 2022; Xie et al., 2023). Recently, researchers have explored using large language models in evaluation metrics, such as GPTScore (Fu et al., 2023), GEMBA (Kocmi & Federmann, 2023), and InstructScore (Xu et al., 2023b). They also benchmark foundation models by prompting LLMs with carefully designed instructions (Bai et al., 2023; Zheng et al., 2023). However, there are several limitations when using LLMs as evaluators, such as position bias and verbosity bias (Zheng et al., 2023). Moreover, Chang et al. (2023) found

that LLMs are heavily affected by the contamination problem, making them perform much better on memorized text than non-memorized text on many downstream tasks.

**Human Evaluation for Stories** Human evaluation is also used to evaluate different aspects of story quality, such as coherence (Xu et al., 2018; Peng et al., 2018), relevance (Yang et al., 2023; 2022; Jhamtani & Berg-Kirkpatrick, 2020), interestingness (Bae et al., 2021) and so on. To comprehensively cover all aspects, Chhun et al. (2022) suggested 6 human criteria for the story: relevance, coherence, empathy, surprise, engagement, and complexity. However, they showed that the inter-annotator agreement of human evaluation on these subjective aspects is low. Karpinska et al. (2021) also highlighted the perils of crowdsourced human judgments from Amazon Mechanical Turk due to under-qualified workers and lacking reproducibility details.

**Personalization in Natural Language Processing** Personalization has been well studied in many recommendation systems (Das et al., 2007; Xu et al., 2022) and search applications (Croft et al., 2001; Shi et al., 2023). Recently, researchers have also highlighted its importance in natural language processing (Flek, 2020; Dudy et al., 2021). Several recent studies have investigated LLMs' capabilities in capturing personalization (Chen et al., 2023; Kang et al., 2023; Salemi et al., 2023) or prompting for personalized recommendations (Lyu et al., 2023; Chen, 2023; Li et al., 2023). In this paper, we incorporate personalization into the evaluation, customizing it according to individual preferences.

## 3 PERSONALIZED STORY EVALUATION DATASET

There are two main challenges in constructing personalized story evaluation datasets. First, it is difficult to collect preference labels for long stories. It requires the reviewer to identify their preferences first and read the whole story to provide a review. This process is time-consuming and costly. We can also use the existing reviews dataset such as the movie dataset IMDB [1]. However, the second challenge is these online movies and reviews have been implicitly exposed to the training phase of many LLMs. This contamination problem may lead to evaluation bias for LLM-based evaluation models. We investigate how the contamination problem affects the evaluation of GPT-4 and LLaMA-2 in Appendix A.1.2. Overall, for LLM-based evaluation, contamination leads to unfair high rating on exposed plots, compared to unexposed ones.

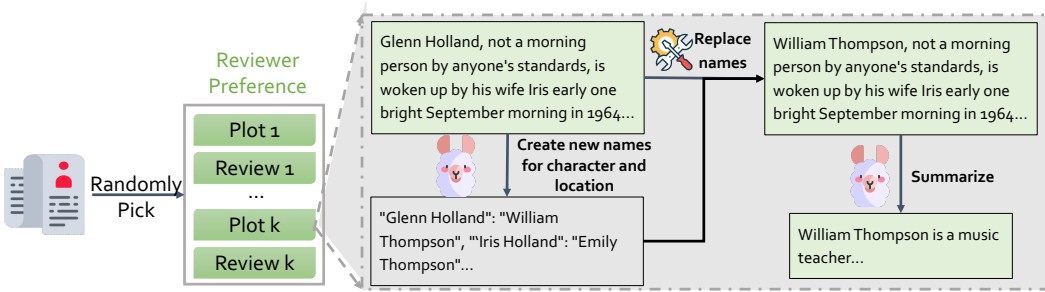

Figure 2: The flowchart to construct our dataset. We use oasst-30b (Köpf et al., 2023), an instruction-tuned LLaMA-based model for anonymization and summarization. The prompts are listed in Figure 9.

To address these problems, we create a less biased personalized story evaluation dataset by anonymization of famous characters and summarization from existing plots, as elaborated below:

**Data Construction**. We demonstrate our pipeline in Figure 2. For each reviewer, we first randomly pick several examples from this reviewer's prior reviews [2]. For each plot, if it is already published online, we rewrite it to avoid contamination. Specifically, we use oasst-30b (Köpf et al., 2023) to anonymize and summarize the plots. It is a 30B LLaMA-based model finetuned on OpenAssistant Conversations for alignment. The anonymization makes the character and location names less identifiable and the summarization avoids the text-level memorization while keeping the main idea of the plot. The anonymization is two-step: it first creates the name mapping and then replaces the name. It ensures that the model will not hallucinate new content during the name replacement. In

---

[1] https://developer.imdb.com/non-commercial-datasets/
[2] We assume that the reviewer's preferences are consistent within the review time frame.

Figure 3, we investigate how the anonymization and summarization affect the evaluation performance. LLMs achieved a high correlation with human ratings in original plots, but the performance degraded after anonymization and summarization. Although the main plots remain the same, with only slight differences in recognizable details, it greatly affected the results. It indicates that these techniques can effectively alleviate the memorization problem. More analysis can be found in Appendix A.1.2.

**Preference labels**. Note that we do not have access to personal profiles that directly describes the story genres reviewers would like. Instead, we use existing reviews from the reviewer as the preference labels that typically reflect the evaluation principles and practices. For example, given the reviews in Figure 1, we can infer that Reviewer 1 favors happy endings while Reviewer 2 cares more about the plot complexity.

Finally, we repurpose two personalized story datasets: **Per-MPST** and **Per-DOC**.

**Per-MPST** We modify the movie review dataset MPST (Kar et al., 2020; 2018) for personalization. Each review includes a review text and a score from 1 (lowest) to 10 (highest). The original dataset includes roughly 15K movies and 1 million reviews. We randomly sample 1k movies and use reviews for these movies. As described above, we anonymize the character and location names in the raw story and summarize it. We then group reviews by reviewer ID and remove reviewers that have fewer than 6 reviews. We split the training and validation set based on reviewer IDs to ensure there is no overlap. For each reviewer, we randomly sample $n = 10$ times from their reviews, with $k = 1$ to 5 plot-review pairs as the preference and one plot as the query. Due to the limited context length of LLaMA-2, we limit the maximum length of the prompt to 2500 words (about 4k tokens).

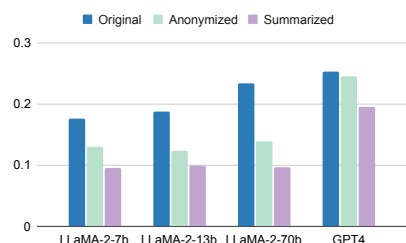

Figure 3: Kendall correlation between the LLM's personalized prediction on movie scores with human ratings. Personalized predictions of all LLMs are also affected by memorization.

**Per-DOC** For the second dataset, we use human evaluation results on novel stories from Yang et al. (2023). There are 7000 unique examples from 403 annotators. Each example consists of two plots generated from the same premise. The annotators were asked to answer various questions and choose their preferred plot for each question. We derive five subjective aspects from the original questions: `Interestingness` (I), `Adaptability` (A), `Surprise` (S), `Character Development` (C), and `Ending` (E). `Interestingness` focuses on the appeal of the overall narrative; `Surprise` indicates unexpected elements or twists in the plot; `Character development` evaluates the emotional and personal connection between characters and events; `Ending` is about satisfaction or appreciation of the ending, and `Adaptability` measures the probability of further developing the story. [3] We use the worker ID to cluster the annotations. Similarly, we split the training and validation set based on reviewers and removed annotators with fewer than 2 annotations. We sample $n = 50$ examples for each remaining annotator and keep $k = 1$ due to the length limitation.

Table 1: Statistics of Per-MPST and Per-DOC. Length is the number of words in the instruction, which includes the instruction template, reviewer preference, and plot query. **I**, **A**, **S**, **C**, and **E** stand for `Interestingness`, `Adaptability`, `Surprise`, `Character Development`, and `Ending`. $k$ is the number of reviews; we fix $k = 1$ for Per-DOC due to the length.

| | | **Per-MPST** | | | | | **Per-DOC** ($k = 1$) | | | | |
|---|---|---|---|---|---|---|---|---|---|---|---|
| | | k=1 | k=2 | k=3 | k=4 | k=5 | I | A | S | C | E |
| **Train** | # Reviewers | 1412 | 1394 | 1385 | 1369 | 1336 | 172 | 171 | 156 | 160 | 155 |
| | # Example | 13254 | 13940 | 13794 | 13480 | 12041 | 1985 | 1856 | 1722 | 1785 | 1574 |
| | Avg. Length | 868.9 | 1235.2 | 1600.3 | 1964.0 | 2123.3 | 2410.9 | 2413.7 | 2411.7 | 2409.8 | 2409.6 |
| **Valid** | # Reviewers | 92 | 92 | 92 | 92 | 92 | 18 | 18 | 15 | 18 | 15 |
| | # Example | 915 | 920 | 920 | 906 | 833 | 234 | 224 | 161 | 162 | 173 |
| | Avg. Length | 857.9 | 1237.1 | 1597.2 | 1956.1 | 2108.4 | 2402.9 | 2399.2 | 2408.4 | 2421.4 | 2404.3 |

---

[3]The corresponding questions are listed in Appendix.

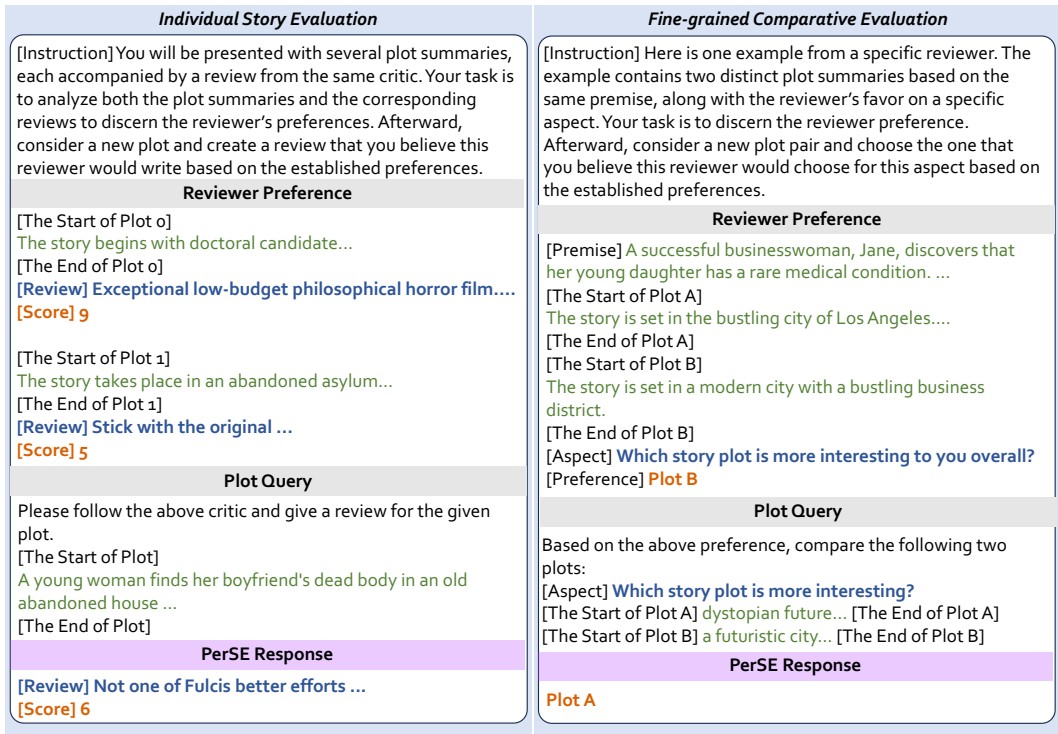

Figure 4: **PERSE**. The reviewer's preference is inferred from their prior reviews. For individual evaluation, the model provides a personalized score as well as a detailed review as an explanation. For the fine-grained comparison, it predicts the reviewer's preference for specific aspects. The plots are in green, the detailed review and fine-grained aspects are in blue, and the review scores are in orange.

## 4 PERSE: PERSONALIZED STORY EVALUATION MODEL

To infer the implicit preference from reviews, we take advantage of the reasoning and instruction-following capabilities of LLMs. We propose an LLM-based personalized story evaluation model (**PERSE**) to generate a personalized review for a specific reviewer. It is finetuned on several instruction data to understanding the reviewer's preferences from the limited prior reviews and follow the instruction to give a subjective review from this reviewer's perspective.

**Problem Formulation** Given a story $x$ and a reviewer $u$, the evaluation model M predicts a personalized review $y = \{y_1, \cdots, y_T\}$ of the story according to the preference of this reviewer $c_u$. $T$ is the length of the review. The review can be a numerical score for an individual plot, or a comparative evaluation between two plots. It can also include a textual explanation for the decision. The preference is defined as a history of this reviewer's reviews on other stories: $c_u = \{(x_{u_1}, y_{u_1}, a_{u_1}), \cdots, (x_{u_k}, y_{u_k}, a_{u_k})\}$, where $k$ is the number of plot-review pairs in the history. $x_{u_i}$ and $y_{u_i}$ are the $i$-th story and review of reviewer $u$, and $a_{u_i}$ is the textual description of the review. It can be the detailed review text or the aspect that this review focuses on.

As demonstrated in Figure 4, **PERSE** can provide a personalized review for both individual and comparative evaluation. It first infers the reviewer's preferences based on the given examples, and then provides a specific review for the new plot or plots. For the individual scoring evaluation, $a_{u_i}$ is the review text describing the detailed reason for the score and $y$ is a score from 1 to 10, which is also treated as a text token in the LLM output. However, instead of directly predicting the numerical score $y$, **PERSE** generates the review content before giving the score, akin to chain-of-thought (Wei et al., 2022). Therefore, for the individual evaluation model **PERSE**$_\text{ind}$, the goal is $y_\text{ind} = \{a, y\} = M_\text{ind}(x, c_u)$. For the comparative setting **PERSE**$_\text{comp}$, the model compares two plots based on the five fine-grained aspects in Per-DOC. $y$ is a textual description of the choice and $a$ is the aspect. We use one unified model for all aspects by adding the specific aspect as the input, which is represented as $y_\text{comp} = M_\text{ind}(x, c_u, a)$.

As described in Section 3, we split the dataset based on reviewers to avoid overlap between training and inference. This ensures that the model only has access to the reviews selected at inference time for a given reviewer, rather than memorizing them during training. We use instruction-tuning to help **PERSE** better reason about the reviewer's preference. In particular, we tune LLaMA-2 on the same training objective for both the individual and comparative settings: $L = -\sum_{t=0}^{T} \log p(y_t|\rho(\boldsymbol{x}, \boldsymbol{c}_u, \boldsymbol{a}), \boldsymbol{y}_{<t})$, where $\rho$ is a template function (black text in Figure 4) mapping the reviewer's preference, story query, and the optional aspect into a single textual instruction.

## 5 EXPERIMENT

### 5.1 EXPERIMENTAL SETTING

We implement **PERSE** based on LLaMA-7b-chat and LLaMA-13b-chat, tuning them on Per-MPST and Per-DOC. In our main experiments, we use $k = 3$ for Per-MPST and $k = 1$ for Per-DOC. Each model in our experiments was trained on 8 x 80G A100 GPU with a learning rate of 1e-5. We set the batch size to 4 for **PERSE**-7b and 2 for **PERSE**-13b. **PERSE**$_{\text{ind}}$-7b and **PERSE**$_{\text{ind}}$-13b converged after 2k/6k steps on Per-MPST respectively. We trained two unified models on Per-DOC for all aspects. Full training details are in Appendix A.2. For inference, we set the temperature to 0.8 and limit the maximum generation length to 600. All reviewers in the test set are new reviewers that **PERSE** never sees during the training phase. We report Pearson, Spearman, and Kendall-Tau correlation coefficients to measure the agreement between human scores and **PERSE** scores for individual story evaluation. For comparative evaluation, we view each aspect as a binary classification between two plots and report the accuracy as the main metric.

**Baseline** We run a simple baseline that directly uses the average scores from given prior reviews as the prediction. It is calculated as $y = \frac{1}{k}\sum_{i=0}^{k} y_{u_i}$ on Per-MPST. For Per-DOC, since we only have one comparison in the instruction ($k = 1$), we directly use this answer as the output. On Per-MPST, we add an additional baseline matrix factorization (MF) (Koren et al., 2009), which is commonly used in the recommendation systems. However, on Per-DOC, both plot pairs and the annotators of the test set have no overlapping with the training set, so the matrix factorization cannot apply to this setting. We also evaluate several LLMs, including the pre-trained LLaMA-2-chat from 7b to 70b and GPT-4, with the same prompts and generation configurations.

### 5.2 MAIN RESULTS

We report the performance of the individual personalized evaluation model **PERSE**$_{\text{ind}}$ on Per-MPST and the fine-grained comparative evaluation model **PERSE**$_{\text{comp}}$ on Per-DOC.

**Individual Personalized Evaluation** As shown in Table 2, **PERSE**$_{\text{ind}}$-13b significantly outperforms all baselines on correlations with unseen reviewers, and **PERSE**$_{\text{ind}}$-7b is comparable to GPT-4. In particular, **PERSE**$_{\text{ind}}$-13b achieves a typical high 0.345 Pearson correlation between its predictions and human scores, indicating that our model effectively captures the reviewer's preference from the given reviews. On the other hand, the results show that it is difficult for LLMs to directly infer the reviewer's preference without instruction-tuning. All LLaMA-2 baselines underperform the trivial baseline which just uses the average score from the review history. This observation is consistent with Kang et al. (2023) who show that pre-trained LLMs struggle to understand reviewers' preferences

Table 2: Pearson, Spearman and Kendall correlations of each method with human ratings on Per-MPST. We use three reviews ($k = 3$) to represent user preferences. All results have a p-value less than 0.05. **PERSE**-7b is comparable to GPT-4 and **PERSE**-13b significantly outperforms GPT-4.

|  | Pearson | Spearman | Kendall |
|---|---|---|---|
| Simple Baseline | 0.301 | 0.302 | 0.230 |
| MF | 0.308 | 0.313 | 0.269 |
| LLaMA-2-7b | 0.146 | 0.117 | 0.094 |
| LLaMA-2-13b | 0.172 | 0.182 | 0.147 |
| LLaMA-2-70b | 0.214 | 0.232 | 0.181 |
| GPT-4 | 0.315 | 0.312 | 0.253 |
| **PERSE**$_{\text{ind}}$-7b | 0.307 | 0.329 | 0.263 |
| **PERSE**$_{\text{ind}}$-13b | **0.345** | **0.368** | **0.293** |

and use them for a personalized score. We believe one possible reason is that both the pre-training phase and reinforcement learning from human feedback (Ouyang et al., 2022) are aligning the model towards more objective and common human values, hindering personalization. This is consistent with Kirk et al. (2023) who claim that the aggregate fine-tuning process may not well represent all

human preferences and values. However, we observe that with targeted instruction-tuning on only a few training data, LLMs can effectively infer personalized preferences and align with them.

**Fine-Grained Comparative Evaluation** We present the accuracy in Table 3. We merged all aspects and jointly trained two unified models by adding aspect tags to the instruction. Our $\textbf{PERSE}_{\text{comp}}$ achieves the best performance on all aspects. Compared to $\textbf{PERSE}_{\text{comp}}$-13b, $\textbf{PERSE}_{\text{comp}}$-7b achieves comparable performance on `Surprise` but lags behind on other aspects. For baselines, the pre-trained LLaMA only achieved comparable performance with the simple baseline, with around 50% accuracy on most aspects. One possible reason for the poor performance is that we only have $k = 1$ review due to the context length limitation, making it more difficult to capture the preference. Even so, our method also only gets k=1, but we're doing great. Meanwhile, GPT-4 does better in capturing `Surprise` than other LLM baselines, but still does not show advantages in other aspects.

Table 3: Fine-grained prediction accuracy on Per-DOC with $k = 1$. $\textbf{PERSE}$-7b and $\textbf{PERSE}$-13b were trained on all aspects. $\textbf{PERSE}$ outperforms all baselines on all aspects. The p-value for t-test are smaller than 0.05.

|  | Interestingness | Adaptability | Surprise | Character | Ending | Average |
|---|---|---|---|---|---|---|
| Simple Baseline | 0.466 | 0.478 | 0.460 | 0.469 | 0.515 | 0.477 |
| LLaMA-2-7b | 0.466 | 0.491 | 0.453 | 0.481 | 0.503 | 0.479 |
| LLaMA-2-13b | 0.422 | 0.451 | 0.477 | 0.481 | 0.517 | 0.470 |
| LLaMA-2-70b | 0.517 | 0.507 | 0.431 | 0.505 | 0.545 | 0.501 |
| GPT-4 | 0.502 | 0.496 | 0.596 | 0.506 | 0.543 | 0.529 |
| $\textbf{PERSE}_{\text{comp}}$-7b | 0.572 | 0.565 | **0.619** | 0.565 | 0.560 | 0.576 |
| $\textbf{PERSE}_{\text{comp}}$-13b | **0.621** | **0.570** | 0.616 | **0.607** | **0.597** | **0.602** |

## 5.3 ANALYSIS

Here, we show some additional experiments to investigate the modeling of personalization in LLMs, and list several observations. More experiments are in Appendix A.3.

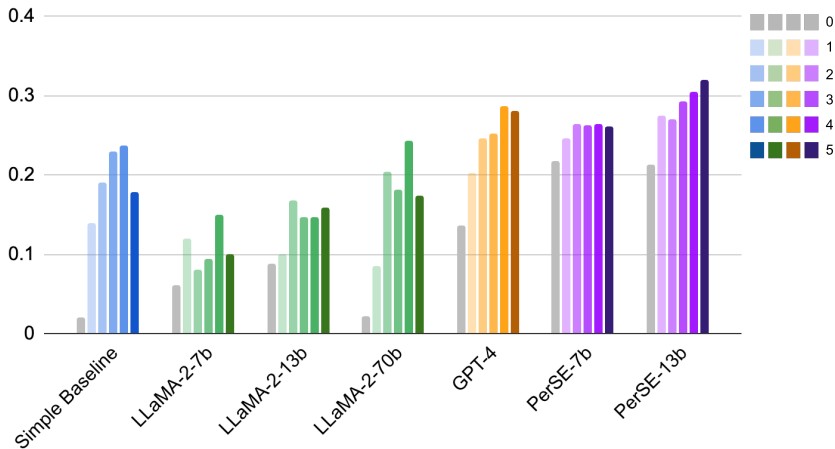

Figure 5: Kendall correlation on Per-MPST with different numbers of reviews ($k$) in reviewer history. Having more reviews benefits $\textbf{PERSE}$-13b, but the increased complexity may harm the performance of LLaMA.

**PERSE achieves a higher correlation with more reviews.** We explore how many reviews are required to establish the reviewer's preference in Figure 5. For $\textbf{PERSE}$-7b and $\textbf{PERSE}$-13b, we train the models on different subsets of Per-MPST as shown in Table 1. $k = 0$ indicates that there is no personalized examples in the instruction, which is a baseline for story evaluation without personalization. We randomly selected a score between 1 to 10 for the simple baseline for $k = 0$. The poor performance on $k = 0$ for all baselines suggests that an overall score does not work for story evaluation. Even for $\textbf{PERSE}$ which has already been finetuned on Per-MPST, it is still not enough to have satisfactory results in evaluation, highlighting the importance of personalization in story evaluation. When we increase the number of reviews, it is easier for $\textbf{PERSE}$-13b to capture

the reviewer's preference. However, for weaker baselines such as pre-trained LLaMA-2, they fail to benefit from more reviews. Furthermore, the simple average baselines also drop after 4 reviews. This indicates that although more reviews provide more information about the reviewer, it also increases the complexity of the context and may introduce noise. Therefore, if not limited by the context length, we suspect that the performance of **PERSE**-13b will also drop after achieving its maximum capability of inferring from complicated context with potential noise.

**More reviews improve the robustness of PERSE.** Previous studies have shown that large language models are sensitive to the example order (Lu et al., 2022). Moreover, the assumption that the preference is constant during these reviews may not hold in a real scenario. Therefore we randomly shuffle the reviews and test for three times to mitigate the potential influence of example order. We demonstrate the average performance with lines and the standard deviation by shadow regions in Figure 6. We can see that **PERSE**-13b stably outperforms other baselines on average. Furthermore, more reviews increase the robustness of **PERSE** to the order change, indicated by a smaller shadow region. It shows **PERSE** has successfully captured the implicit reviewer's preference from these reviews. In contrast, the pre-trained LLaMA-2 are still very sensitive to the order, with a larger variance shadow.

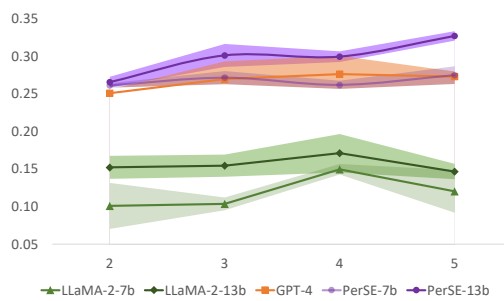

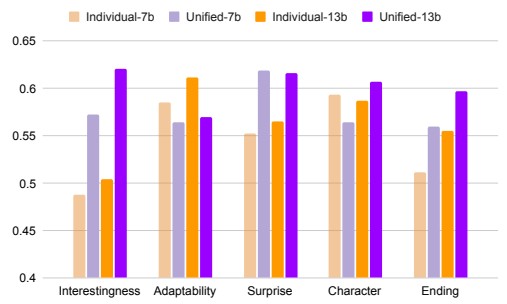

Figure 6: Kendall correlation on Per-MPST with different orders of reviews. The shadow indicates the variance while the line is the average performance among three trials. **PERSE** is more stable than baselines.

Figure 7: Accuracy of the unified models and individual models on Per-DOC. The unified models improve performance in most cases.

**Joint training benefits the individual aspects.** We investigate the influence of joint training of different aspects on Per-DOC by training an individual model on each aspect and comparing the performance. As illustrated in Figure 7, the performance in most aspects is enhanced by the joint training, where the models are exposed to more data, i.e., different aspects can benefit each other. For example, the performance of capturing `Interestingness` and textttSurprise, and evaluation of the quality of `Ending` are weaker under the individual setting, but are enhanced by other aspects during the joint training, resulting in significant improvement. For those separate models, they are better at capturing the preference for `Adaptability` and `Character Development`. We hypothesize that these two aspects are related to the setting of the plot, which is more structured. This may lead to a clearer preference that is easier to capture with single-aspect data.

**GPT-4 tends to be nice and general.** In Figure 8 we show an example from Per-MPST. From the annotated reviews, we can see that this reviewer is critical about the plots, and especially cares about the novelty. However, even given this reviewer's preference, GPT-4 predicts a positive review, which may be caused by alignment towards safety or harmlessness. LLaMA-2-70b is stricter but still gives a score of 4. **PERSE** cares more about the steady terribleness and only gives 3, which is more consistent with this reviewer's true score. Moreover, from the review preference, we find that, unlike most people, this reviewer does not pay much attention to complicated themes. However, GPT-4's "one-size-fits-all" evaluation offers a high score for this theme. **PERSE** cares more about the visual preference of this reviewer, giving a more reviewer-specific rating. This indicates that **PERSE** can better evaluate stories based on personalized preferences rather than a general and nice evaluation principle without any personalized preference.

| Review Preference |
|---|
| [The Start of Plot 0]
A newscast is shown about a year after a prehistoric piranha attack on Lake Victoria. The lake is now uninhabitable, and the town has been abandoned due to the drying up of the main revenue source, tourism...
[The End of Plot 0]
[Review] Quote The Hoff: Welcome to rock-bottom.. features everything that gives horror , as well as sequels in general , a bad reputation [Score] 3
[The Start of Plot 1]
Evelyn Summers is a shy and unattractive but intelligent girl who likes witchcraft and has a crush on Bill Natolly, her handsome English teacher...
[The End of Plot 1]
[Review] honey, I think we should see other people... Dont you just hate it when this happens? Now that you finally found yourself a gorgeous looking high school sweetheart, she turns out to be a murderous witch, capable of making you eat glass beer bottles with the touch of her hand! [Score] 5 |
| **Plot Query** |
| Professor Jack Hanson and his colleagues go to the Gibraltar ice shelf to take ice core samples and discover evidence of global warming via a presentation to the UN that is poorly received. However, when multiple buoys in the North Atlantic show a sharp drop in ocean temperature, Professor Christopher Rapson believes Jack's theories and works with him to build a forecast model....
At the US refugee camp in Austinia, President Elizabeth Blake dies when her motorcade is caught in the superstorm, and Vice President Elizabeth James Becker becomes the new president. The next morning, Sam's group realizes the storm is over, and they begin their journey to the evacuation center. |
| **Reference** |
| [Review] I mean, you can't judge a film negatively because its far fetched. The whole thing may be scientifically unjustified, but it has to remain a little appealing to the audience, no? The dreadfulness of this film lies in the same old, disastrous cliche that keep on getting repeated over and over again. [Score] 2 |
| **GPT4 Response** |
| [Review] This film offers a chilling, literally and metaphorically, journey into the devastating effects of global warming, a pressing issue that's often brushed aside... Nonetheless, the film effectively uses the fear of the unknown future of our planet to craft an engaging narrative. [Score] 7 |
| **LLaMA-2-70b Response** |
| [Review] I'm just not sure if it all adds up to a compelling story. I mean, the characters are pretty one-dimensional, and the plot feels like it's all over the place. [Score] 4 |
| **PerSE Response** |
| [Review] I really am amazed at how steadily and consistently terrible the movies are. Oh well, at least the disaster itself was kind of cool to watch, although even that had its flaws and clearly visible budget restrictions. Too bad they didn't spend a little more money on it, and actually make it a decent movie. [Score] 3 |

Figure 8: An example for evaluating the individual story from the given reviewer's preference. The reference is the ground-truth reviews given by this reviewer. Some content has been abridged due to space considerations. More cases are shown in the Appendix.

# 6 CONCLUSION AND DISCUSSION

In this paper we focus on the personalized evaluation of open-ended generation. We investigate several methods to create unbiased personalization datasets from existing corpora, helping to overcome the exposure of LLM evaluation models to pre-existing datasets. We also re-purpose two story evaluation datasets Per-MPST and Per-DOC for comprehensive personalized evaluation under different scenarios. Moreover, we proposed **PERSE** for personalized story evaluation. It achieves a high correlation with the reviewer's judgment, outperforming GPT-4 in both the general judgment and fine-grained pairwise comparison.

While this research makes notable strides in addressing the challenge of personalized story evaluation, it is not without its limitations. Primarily, **PERSE** uses reviews from reviewers to capture their preferences, assuming constant preference over time that may not mirror real-world scenarios. Additionally, the current context length of large language models limits the number of reviews, which might affect the comprehensive understanding of a reviewer's preference. However, with the development of large language models with longer context windows, we believe that more reviews can be utilized for better modeling of the reviewer's preference.

This personalized story evaluation also encourages personalized story generation. It can be used to align the existing story generation models to a specific reviewer, making it more efficient and suitable for the individual taste and preference. Moving forward, we aim to use **PERSE** to further improve the alignment between story generation models and the distinctive requirements of reviewers.

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

# A  APPENDIX

## A.1  MORE DETAILS ABOUT PERSONALIZATION DATASET

### A.1.1  DATA PROCESSING

In Figure 9 we list prompts we used in Section A.1.2. We anonymize the raw plot by asking LLMs to identify characters and local names and create new names for them. Based on the JSON mapping it generates, we replace those names with new names. We do not directly ask LLMs to replace names because they sometimes hallucinate new plots during the replacement. For the characters with the same family names, LLMs are able to create new character names that still have the same last names (but not the same as the original last names). For example, 'Glenn Holland' and 'Iris Holland' are mapped to 'William Thompson' and 'Emily Thompson'.

For Per-DOC, we define five aspects based on the questions in Yang et al. (2023):

1. `Interestingness`: Which story plot is more interesting to you?
2. `Adaptability`: In your opinion, which one of the plots above could generate a more interesting book or movie (when a full story is written based on it)?
3. `Surprise`: Which story plot created more suspense and surprise?
4. `Character Development`: Which story's characters or events do you identify with or care for more?
5. `Ending`: Which story has a better ending?

These aspects evaluate the three key elements in the story: Interestingness and Surprise for the plot, Character development for the character, and Ending and Adaptability for the setting. For each question, there are four options: plot A, plot B, both are good, neither is good. We remove the examples with the answer of 'Both' and 'Neither' because they do not show preference.

We illustrate the length distribution of the movie plot in Per-MPST and the story in Per-DOC in Figure 10b and 10c. For Per-MPST, we also provide the length distribution of the raw plots in Figure 10a.

| | |
|---|---|
| **Anonymization** | Here is one plot:
**{plot}**
Please create a JSON mapping of current character and location names to new, distinctive names. In this mapping, the current names will act as keys and the new names as values. For instance, if you were to change the name 'Diego' to 'Sherry Evans', the corresponding JSON entry would be: {{'Diego': 'Sherry Evans'}}. The task requires you to replace all character and location names in the text with alternative names, and then provide the mapping relationship as a JSON object. |
| **Summarization** | Provided below is a narrative:
**{plot}**
Kindly analyze this story and provide a clear and succinct summary of the key events. |
| **Individual Story Evaluation** | Here we have one plot. Please give a score for 1 to 10 for the following plot, where 1 is the lowest and 10 is the highest. If you already know the plot, give the name. But remember do not depend on any public review score you already remember.
[Plot] **{plot}**
Please only reply a JSON-format with the following keys: "Score", "Title". If you cannot identify the title, respond with "N/A" for that field. |
| **Pairwise Story Evaluation** | Here we have two plots: plot1 and plot2. Please based on the description to choose which one is better and give your reasons. If you know the movie title of this plot, please tell me the titles as well.
[Plot1] **{plot1}**
[Plot2] **{plot2}**
Please only reply a JSON-format with the following keys: "Choice", "Reason", "Plot1 Title", "Plot2 Title". If you cannot identify the title, respond with "N/A" for that field. |

Figure 9: Prompts used in Section A.1.2. The blue text is the placeholder for plots.

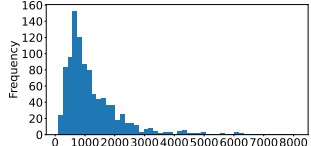
(a) Raw movie length in MPST v2.

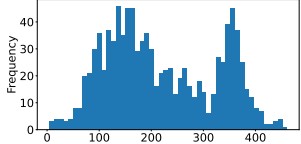
(b) Movie length in Per-MPST.

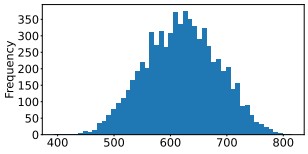
(c) Story length in Per-DOC.

Figure 10: Length Distribution of Per-MPST and Per-DOC. The x-axis is the length and the y-axis is the frequency.

Table 4: GPT-4 in comparing two movies. The plot with the correct predicted title is viewed as a known plot by GPT-4. Cons. is the percentage of consistent results when swapping the order. Bias First is the percentage where GPT-4 favors the first answer more than the ground truth. Percent is the percentage of each story type (raw/anonymized/summarized) recognized as 'both known', 'one known', or 'neither known'. Overall, memorization leads to greater position bias and lower consistency.

|  |  | Accu. ↑ | Cons. ↑ | Bias First ↓ | Percent |
|---|---|---|---|---|---|
| **Both Known** | Raw | 0.714 | 63.0% | 16.5% | 91.0% |
|  | Anonymized | 0.712 | 60.7% | 17.8% | 73.0% |
|  | Summarized | 0.753 | 73.4% | 12.9% | 42.5% |
| **One Known** | Raw | 0.778 | 78.9% | -11.1% | 9.0% |
|  | Anonymized | 0.804 | 71.7% | -6.5% | 23.0% |
|  | Summarized | 0.632 | 82.4% | 1.5% | 34.0% |
| **Neither Known** | Raw | / | / | / | 0.0% |
|  | Anonymized | 0.500 | 62.5% | 25.0% | 4.0% |
|  | Summarized | 0.660 | 85.1% | 4.3% | 23.5% |

### A.1.2 CONTAMINATION IN EVALUATING EXISTING STORIES

Many online stories have been exposed to LLMs during pretraining, which may lead to bias during evaluation. We first investigate how such contamination affects LLMs when evaluating general stories. We use average movie ratings from the non-commercial IMDB dataset,[4] containing plots evaluated by thousands of reviewers with scores ranging from 1 to 10. We let GPT-4 to evaluate the movie plot and ask it to identify the movie title to check the memorization.

We consider a movie to be "known" by GPT-4 if the title is correct, and split the results into three groups based on memorization status: GPT-4 knows both plots, knows one, or knows neither. We calculated prediction accuracy (Accu.), consistency (Cons.) and bias first within each group. Consistency measures how many judgments are consistent after changing the order in which the two plots are presented. Bias first is defined as inappropriate preference for the first one. It is calculated by subtracting the percentage where GPT-4 favors the first plot by the true percentage of the first. We put the prompts used for investigation on memorization in Figure 9.

We investigate the memorization problem in two settings: the individual evaluation is to predict a score (1 to 10) for a single story, and the pairwise story evaluation is to compare two plots.

**Pairwise story evaluation** We create 200 movie pairs, where each pair consists of two movie plots whose rating differ by 1 point. We ask GPT-4 to identify the titles and then conduct pairwise comparison [5]. Results on the original IMDB movie plots are reported in the 'Raw' rows of Table 4. We can see that GPT-4 knows at least one of movies in the pair. Moreover, if GPT-4 knows exactly one of the two plots, it is more consistent in its judgment and has a lower position bias. We find it is because GPT-4 tends to choose the known plot. To alleviate the effect of memorization, we ask GPT-4 to identify the characters and local names in the plot and randomly replace them with similar names, ('Anonymized' in Table 4); doing so reduces the percentage of both known pairs by 18%. However, 96% of pairs still have at least one known plot. Therefore, we further summarize the

---

[4]`https://developer.imdb.com/non-commercial-datasets/`
[5]We used the `gpt-4-0613` version from `https://openai.com/gpt-4` with default settings.

anonymized plot ('Summarized'), reducing both known to 42.5% and increasing neither known to 23.5%. In all three groups, the summarized plots have the highest consistency and lowest position bias. Moreover, compared to the other two groups, the neither known group exhibits much lower accuracy despite keeping the main plot points, indicating that memorization can result in misleadingly high performance in story evaluation.

We further calculate the 'Bias Known' on the 'One known' group by subtracting the percentage that GPT-4 favors the known plot by the true percentage where this plot is better. In Table 5, we can see that for all raw, anonymized, and summarized plots, GPT-4 has an obvious tendency for the known plot when it can identify one of the plot pair. This tendency is more obvious in the summarized plots. We suppose it is because, with the data processing, the uncertainty of the prediction increases. It makes the model more conservative, believing in what it has known. However, GPT-4 also shows high consistency and low position bias on the 'neither known' group (see Table 4), indicating that when facing two novel stories, it can get rid of the effect of memorization and evaluate based on the plots.

**Individual story evaluation** We also investigate the influence of memorization on individual story evaluation. Similarly, we ask the GPT-4 to identify the movie title and give a score (1 to 10) for this plot. We divide the results into 'Known' and 'Unknown' according to the success of the title identification. We calculate the correlation between the prediction scores and the average scores in IMDB. The results are shown in Table 6. The percentage of known significantly decreases after anonymization and summarization, indicating the effectiveness of alleviating memorization issues. Although the correlation on known plots is very high, it drops after GPT-4 fails to identify the plots. It shows that the memorization issue makes the evaluation of GPT-4 unreliable.

Table 5: Prediction on 'One known' Group in pairwise comparison of GPT-4. The 'Raw', 'Anonymized', and 'Summarized' have the same meaning with Table 4. 'Bias known' is defined as the case that GPT-4 more favors the known plot than the ground-truth.

|  | Bias Known |
|---|---|
| Raw | 0.222 |
| Anonymized | 0.283 |
| Summarized | 0.397 |

Table 6: Performance of GPT-4 in predicting average movie scores. Percent is the percentage of each type of stories (raw/anonymized/summarized) being recognized as 'known', 'Unknown'. Memorization heavily affects performance, but its impact decreases with anonymization and summarization.

|  |  | Pearson | Spearman | Kendall | Percent |
|---|---|---|---|---|---|
| **Known** | Raw | 0.680 | 0.718 | 0.590 | 84.5% |
|  | Anonymized | 0.682 | 0.680 | 0.548 | 57.5% |
|  | Summarized | 0.621 | 0.648 | 0.552 | 27.0% |
| **Unknown** | Raw | 0.460 | 0.470 | 0.364 | 15.5% |
|  | Anonymized | 0.216 | 0.289 | 0.222 | 42.5% |
|  | Summarized | 0.232 | 0.271 | 0.217 | 72.5% |

**Personalized story evaluation** We also explored the influence of memorization in personalized story evaluation for different LLMs. We provided one review from the same reviewer as the few-shot example and asked LLMs to predict a personalized score for the new plot. We conducted the experiment on randomly chosen 400 reviewers of Per-MPST with $k = 1$ and calculated the Kendall correlation between the human ratings and the predicted score in Figure 3 for LLaMA-2 and GPT-4. Similarly, LLMs achieved a high correlation with human ratings in original plots, but the performance degraded after anonymization and summarization. Although the main plots remain the same, with only slight differences in recognizable details, it greatly affected the results. Both experiments highlight that the memorization results in great bias in LLM-based story evaluation models, making them unreliable for both general story evaluation and personalized evaluation.

## A.2 TRAINING DETAILS

We demonstrate the framework of **PERSE** in Figure 11. Each model in our experiments was trained on 8 x 80G A100 GPU with a learning rate of 1e-5. We set the batch size to 4 for **PERSE**-7b and 2 for **PERSE**-13b. **PERSE**$_{ind}$-7b and **PERSE**$_{ind}$-13b converged after 2k/6k steps on Per-MPST respectively. We trained two unified models on Per-DOC for all aspects by finetuning 7b and 13b LLaMA-2-chat. **PERSE**$_{comp}$-7b converged after 1k steps and **PERSE**$_{comp}$-13b converged after 2k steps. It took about 10 hours for these two models. For the ablation study, we also trained one model

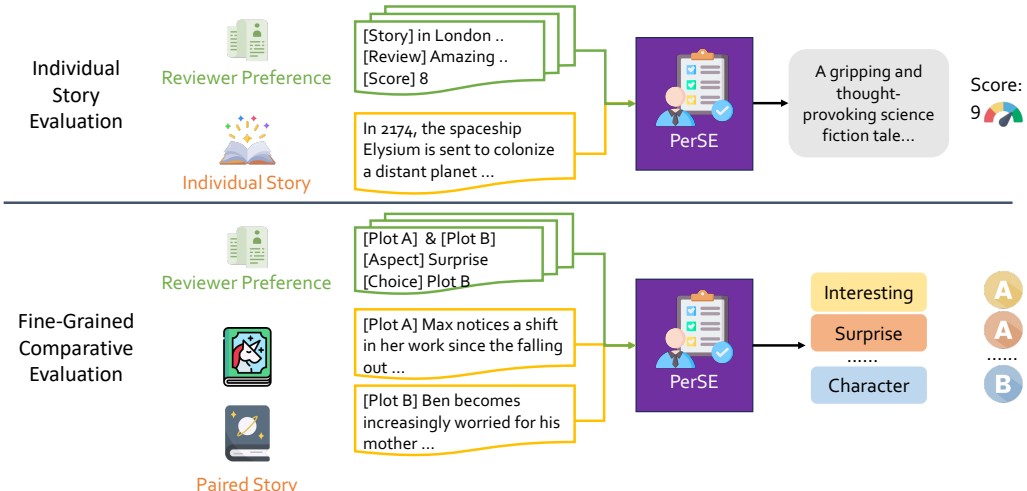

Figure 11: The overall framework of **PERSE**.

Table 7: The ablation study on the review content on Per-MPST. We utilized three reviews ($k = 3$) to represent the reviewer's preferences. The results are the average of three replicate experiments with p-values less than 0.05. Removing review content leads to performance degradation.

|  | **Pearson** | **Spearman** | **Kendall** |
|---|---|---|---|
| LLaMA-2-7b | 0.123 | 0.130 | 0.104 |
| w/o content | -0.007 | -0.002 | -0.005 |
| LLaMA-2-13b | 0.163 | 0.191 | 0.154 |
| w/o content | -0.039 | -0.035 | -0.028 |
| **PERSE**-7b | 0.322 | 0.340 | 0.272 |
| w/o content | -0.088 | -0.064 | -0.057 |
| **PERSE**-13b | 0.381 | 0.378 | 0.301 |
| w/o content | -0.088 | -0.069 | -0.057 |

for each aspect on Per-DOC and each model converged after 500 steps for 7b and 2k steps for 13b. The total training time was around 5 x 5 hours. We plot the curve of the training loss in Figure 12.

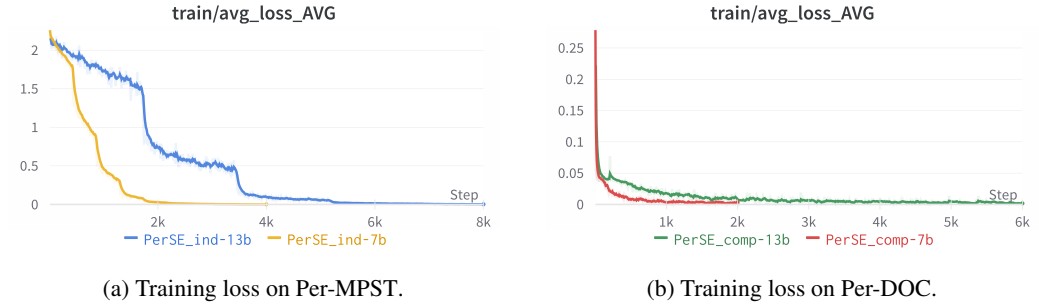

(a) Training loss on Per-MPST.          (b) Training loss on Per-DOC.

Figure 12: Average training loss

## A.3    MORE ANALYSIS

**Chain-of-thought in individual Evaluation** We also investigate the role of review text in **PERSE**. We removed the review text $a_{u_i}$ in the review history $c_u$ and retrained the model. The results are shown in Table 7. We can observe that for all models the performance degraded after removing the

| Review Preference |
|---|
| [The Start of Plot 0]
A train arrives at a station in North China in 1950, carrying prisoners who are war criminals...
[The End of Plot 0]
[Review] A western look at the fall of the Manchurian Dynasty and the life of a true survivor. [Score] 8
[The Start of Plot 1]
In a small southern California town, 24-year-old musician Jason Thompson and his 13-year-old brother Michael are disturbed by the mysterious deaths of the town's citizens....
[The End of Plot 1]
[Review] one of the classic horror films that came out during the genres rebirth during the late 80s. Fueled by the success of Halloween! [Score] 9 |

| Plot Query |
|---|
| The Mindhunters are a group of young FBI students who are undergoing training as profilers. At the last minute, they are joined by an outside observer. As they settle down for the evening, they practice their profiling skills on each other.
In the end, Nicole Williams, the group's sole survivor, shoots and kills Vincent Nguyen, who is revealed to be the killer seeking revenge against Sarah Johnson. The group is commended by the FBI for their bravery, and vow to always remember the island and their fallen comrades. |

| Reference |
|---|
| [Review] The movie was neither scary or thrilling. Just another wanna be action film with a bunch of inane sequences and contrived situations. . [Score] 2 |

| GPT4 Response |
|---|
| [Review] A chilling thriller that keeps you on the edge of your seat, guessing till the very end. The premise of FBI trainee profilers finding themselves in the midst of a real-life hunt is a clever and dynamic plot device. It's the interplay of trust and suspicion among the group. [Score] 7 |

| LLaMA-2-70b Response |
|---|
| [Review] A thrilling story of a group of young FBI students who are undergoing training as profilers on a small island off the coast of North Carolina. [Score] 8 |

| PerSE Response |
|---|
| [Review] Profiling or Crystal Ball Reading? It was supposed to give the audience an inside look at how FBI profiling is really done. Instead, what we get is a bunch of young wannabees playing with people lives. I don't think that making a movie about profiling was the right choice here... [Score] 5 |

Figure 13: The score given by the reviewers on the new plot is very different from the comments with annotations. While LLaMA-2-70b and GPT-4 give a more similar score, PᴇʀSE is able to infer the preference and provide a score that is closer to the true score but far away from the annotated scores.

detailed review content. This highlights the importance of incorporating the chain of thought when evaluating the story, especially for zero-shot settings.

**PᴇʀSE infers the preference instead of copying scores from context.** In Figure 13, we show another example on Per-MPST. From the reviews, we can find the reviewer loves horror elements. However, the new plot, and its level of terror is not satisfactory, which makes the reviewer give a low score on it. Both GPT-4 and LLaMA-2-70b emphasize the horror theme and predict a high score for this plot. We suppose that they are affected by the high review scores in the reviewer's preference, ignoring the analysis of the new plot. In contrast, PᴇʀSE focuses on the boring profiling of the plot, which is more similar to what the reviewer cares about. It gives a score of 5, which is different from the existing review scores but close to the real score this review has for this plot.

**PᴇʀSE is able to provide diverse reviews for the same plot based on different preferences.** In Figure 14, we demonstrate the reviews of the same plot from two reviewers A and B with different preferences. We can see that both the reviewer A and B have read the book. Reviewer A is a critical reviewer and has a high standard for good movies, leading to low scores in the annotated reviews. He then gives a score of 2 because of his disappointment with the movie adaptation. In contrast, reviewer B is relatively tolerant and likes to score high. Although the movie is much worse than the book, the reviewer still gives a score of 6. However, GPT-4 and LLaMA-2-70b give similar high scores in both cases, ignoring the reviewer's preference. Instead, PᴇʀSE is able to give personalized scores for different reviewers, predicting 1 for reviewer A and 8 for reviewer B. Although the predicted score of reviewer B is not as close as GPT-4, it illustrates the positive attitude it captures.

**PᴇʀSE achieves better performance on fine-grained comparative evaluation.** We illustrate one example from Per-DOC in Figure 15. PᴇʀSE successfully predicts the preference on 4 out of 5 aspects, while GPT-4 correctly predicts 3 aspects and LLaMA-2-70b only has 2 success. GPT-4 predicts Plot A for all aspects, ignoring the difference between aspects and outputs an overall

| Review A Preference | Review B Preference |
|---|---|
| [The Start of Plot 0]
Zara encounter a man beating a transsexual prostitute. She tries to intervene but is raped and beaten unconscious..... Zara wakes up and is sad that the man who raped her is dead.
[The End of Plot 0]
[Review] Irredeemable! The viewer is taken on a ride through the tunnel of sado-masochistic grotesquerie, beginning at the end and traveling with jolts and stops back to the start. [Score] 1
[The Start of Plot 1]
Henri Ford is a professor at UCLA who is still grieving the loss of his partner Ryan Miller three years earlier....
[The End of Plot 1]
[Review] Miscast! Actually the novel is rather good. And the film does not depart all that much from it, just enough to create questions.! [Score] 5 | [The Start of Plot 0]
Two police officers respond to a car crash, where two people are killed, but one miraculously survives. The story then cuts to the title credits.
[The End of Plot 0]
[Review] Folks, let me tell you, this is one GREAT weird little thriller, not much blood, but a whole lot of creepiness, a great story, and very british. [Score] 10
[The Start of Plot 1]
Angela is a call girl who is terrorized by a man named Frank, who was just released from prison.
[The End of Plot 1]
[Review] The trouble with anthology films like this, is that you're usually gonna get stories that are better then the others. [Score] 7 |

| Plot Query |
|---|
| Richard Hayes is a financial whiz who's about to earn a million dollars through a bonds scheme. He has a wife, Julianna, and daughter, Grace, and is engaged in an affair with Sophia Taylor, a Southern belle gold digger...
After much turmoil, Richard and Sophia are both acquitted of all charges and are allowed to leave the courtroom, where Richard thanks the jury for their decision. |

| Reference A | Reference B |
|---|---|
| [Review] As someone who has both read the novel and seen the film, I have a different take on why the film was such a flop. First, any comparisons between novel and film are purely superficial. [Score] 2 | [Review] I liked the book better. The film pales in comparison unfortunately.  So, while I dont see this is the worst movie (its OK, NOT BAD), its a time-waster, if that. [Score] 6 |

| GPT4 Response | GPT4 Response |
|---|---|
| [Review] A story that confronts the uncomfortable realities of racial disparities, privilege, and the media's role in shaping public opinion; this plot, albeit intense, presents a new way of examining societal issues. [Score] 7 | [Review] While the plot navigates through themes of social classes and race relations, it misses out on delivering the horror and weirdness that I tend to look for in films. Despite its lack of otherworldly elements, the storyline is strong and it delivers some thrilling moments. [Score] 6 |

| LLaMA-2-70b Response | LLaMA-2-70b Response |
|---|---|
| [Review] Overall, this is a well-crafted and engaging novel that will keep you hooked until the very end. [Score] 8 | [Review] This movie is a great example of how our society can be both hypocritical and discriminatory. [Score] 9 |

| PerSE Response | PerSE Response |
|---|---|
| [Review] In short, it is a perfect example of how elites feel about the great unwashed.I could perhaps forgive the film its sins if it were simply incompetent. But it is not incompetent. [Score] 1 | [Review] Its not a happy movie by any means, but it is thought provoking. Wealth, and the lack thereof is the main focus, and how it can distort ones sense of right and wrong...[Score] 8 |

Figure 14: Reviews from two reviewers on the same plot. **PERSE** is able to give personalized scores based on preference.

evaluation. Instead, **PERSE** cares more about the distinctive attribute of each aspect and gives judgment according to the aspect.

| | |
|---|---|
| | [Premise] A lonely man in his seventies befriends a group of street kids who teach him to find meaning in the simplest of joys, and he, in turn, teaches them about a different perspective on life. They rediscover life and its small joys together.
[The Start of Plot A]
Settings The story is set in an urban city with a mix of old and new architecture.
Characters
 Sophie Wong Sophie Wong is 16 years old, a street kid who has been living on the streets since the age of 12, when she ran away from an abusive home.Mark Chen Mark Chen is 25 years old, a caring and compassionate social worker who befriends Edward and the street kids.Edward James Edward James is 75 years old, a retired math teacher, living alone in a small apartment since his wife died three years ago.
Outline
 1. Edward becomes lost in his grief after his wifes death and becomes detached from the world around him.
 2. Sophie and the other street kids discover him sleeping on a park bench one night and, sensing his loneliness, initiate a friendship with him.
 3. Mark, the social worker, recognizes Edwards situation and offers his help, which brings him closer to the street kids and helps him find a new purpose in life. |
| **Reviewer Preference** | [The End of Plot A]
[The Start of Plot B]
Settings The story is set in a small town in the United States.
Characters
 Tito Robles Tito Robles is 15, a street kid who is the leader of the group he befriends John with, and together, they find meaning in life.Jane Davis Jane Davis is 40, Drews wife, and a friendly and welcoming presence in the town.Ben Smith Ben Smith is 45, a retired military man who lives in the same town and provides help and advice to John and the street kids when they need it.John Doe John Doe is 75, a retired man with a small house and a lonely life.Drew Davis Drew Davis is 50, the local bartender and a friend of John, who helps him connect with the street kids and their way of life.
Outline
 1. John becomes friends with Tito and the street kids, and together they rediscover the simple joys of life despite their different ages and backgrounds.
 2. Drew, Jane, Ben, and other townspeople play important roles in helping the group of friends and teaching them about life and caring for one another.
 3. The man decides to help the street kids and provides them with a house filled with toys and games.
[The End of Plot B]
[Interestingness] Plot A [Adaptability] Plot B [Surprise] Plot A [Character Development] Plot A [Ending] Plot A |
| | [Premise] A struggling artist, living in a small town, stumbles upon an antique store that holds a mysterious painting with the power to change the course of her life, but at what cost?
[The Start of Plot A]
Settings The story is set in a small, rural town in the American South.
Characters
 Maddie James Maddie James is 30 years old, Emmas best friend and roommate, with a quirky personality and a passion for art.Charles Carson Charles Carson is 45 years old, Emmas high school art teacher, who saw her potential and pushed her to pursue her artistic ambitions.Emma Watson Emma Watson is 24 years old, with wild, curly hair and big, expressive eyes.
Outline
 1. Emma discovers the mysterious painting at the antique store and starts to experience strange occurences around her town, leading her to suspect the true power of the art work.
 2. Motivated by her desire to understand the paintings power, Emma begins to research and is guided by her art teacher and mentor towards her potential as an artist.
 3. Emma starts to experience success as an artist and is approached by a powerful art dealer who reveals the true nature and power of the mysterious painting and offers her a tempting deal that threatens her family and friends. |
| **Plot Query** | [The End of Plot A]
[The Start of Plot B]
Settings The story is set in a small town surrounded by vast, open fields and rolling hills.
Characters
 Jackson Wrightson Jackson Wrightson is 29 years old, an art appraiser and Elaras ex-boyfriend, who is both supportive and a source of tension in her life.Elara Kassin Elara Kassin is 32 years old, with a kind heart and a struggling artist living in a small town.Lila Williams Lila Williams is 26 years old, Elaras best friend and a supportive companion who helps Elara on her journey to uncover the truth.Iris Beller Iris Beller is 61 years old, a kind and wise antique store owner, who serves as a confidante and mentor to Elara.Adrian Roth Adrian Roth is 33 years old, charming with disheveled hair and a mysterious demeanor, runs an antique store with a secret to hide.
Outline
 1. Elara discovers the mysterious painting at Adrians antique store, but quickly realizes the painting is more than just a simple work of art.
 2. Elara starts to experience strange dreams and visions, causing her to explore the paintings true purpose and the consequences of her involvement in its magic.
 3. Elara, with the help of Lila, Jackson, and Iris, uncovers Adrians true intentions and the dark ritual required to harness the paintings power.
[The End of Plot B] |

| | |
|---|---|
| **Reference** | [Interestingness] Plot B [Adaptability] Plot A [Surprise] Plot A [Character Development] Plot B [Ending] Plot A |
| **GPT-4** | [Interestingness] Plot A [Adaptability] Plot A [Surprise] Plot A [Character Development] Plot A [Ending] Plot A |
| **LLaMA-2-70b** | [Interestingness] Plot A [Adaptability] Plot B [Surprise] Plot A [Character Development] Plot A [Ending] Plot A |
| **PerSE** | [Interestingness] Plot B [Adaptability] Plot A [Surprise] Plot B [Character Development] Plot B [Ending] Plot A |

Figure 15: One case of comparative evaluation on Per-DOC. **PᴇʀSE** is more similar to this reviewer. However, it fails to capture the preference of `Surprise` in this case.

