# OpenReview forum: "Learning Personalized Story Evaluation"
_ICLR.cc/2024/Conference — Submitted to ICLR 2024_

### Official Review · Reviewer_hjpK · 2023-10-19

**Soundness:** 2 fair
**Presentation:** 1 poor
**Contribution:** 2 fair
**Rating:** 5
**Confidence:** 4

**Summary:**

To perform personalized story evaluation, this paper creates two new datasets based on existing datasets. And the authors also proposes a personalized story evaluation model to infer reviewer preferences and provide a personalized evaluation. Experiments results on the constructed datasets verify the effectiveness of the proposed method.

**Strengths:**

1.The experiments are relatively adequate.
2.The citation work is up to date.

**Weaknesses:**

1.The description of the paper is too fragmented and less logical.
2.The dataset construction section is too cursory, it is recommended to refine it and provide a flowchart.
3.The description of evaluation model is too simple and a framework diagram is necessary.
4.The authors need further clarity on the practical implications of the research work as well as specific application scenarios.

**Questions:**

See above.

---

> ### Author Response · Authors · 2023-11-18
> **Thanks for the valuable comments! Here we discuss these concerns as below**
>
> **Q1. The description of the paper is too fragmented and less logical.**
>
> A1. We have reorganized the paper to make it clearer. We refer the reviewer to the general response for more detailed information.
>
> ---
>
> **Q2. Flowchart & a framework diagram**
>
> A2. Thanks for the valuable suggestions! We add a flowchart for the data construction in Figure 2 and demonstrate our PerSE framework in Appendix Figure 11.
>
> ---
>
> **Q3. need further clarity on the practical implications of the research work as well as specific application scenarios.**
> A3. In this paper, we mainly focus on
>
> 1. *introduce two personalized story evaluation datasets.* By elaborating on the data construction, these two datasets are less affected by the contamination issue of LLM-based evaluation models. They can be used as better benchmarks for both training and testing of the personalized story evaluation models. The pipeline can be easily adapted to other tasks.
>
> 2. *propose PerSE to analyze the unseen reviewers’ preferences from their prior reviewers and predict specific reviews from their views*. By using a small amount of data as the instructions, we enhance LLMs’ capabilities of modeling the reviewer’s preference. The experimental results show that PerSE can predict personalized reviews for new users with only 3 prior reviews.
>
> Thanks to this capability, PerSE can be used in **the recommendation system to select the most suitable story** for a new reviewer without finetuning the preference data of this reviewer. For example, given several stories and three reviews from a new user, we can use PerSE to rank them and provide suggestions for this user.
> PerSE can further **tailor story generation models to align with the unique needs and preferences of different users** by using the score of PerSE as the reward.

---

### Official Review · Reviewer_7x1E · 2023-10-29

**Soundness:** 2 fair
**Presentation:** 1 poor
**Contribution:** 2 fair
**Rating:** 5
**Confidence:** 3

**Summary:**

The research introduces PERSE, a model for personalized story
evaluation. It leverages reviewers' preferences and generates
personalized reviews for stories. Two new datasets are created to
address contamination in story evaluation. PERSE outperforms GPT-4,
demonstrating its effectiveness. The study also explores
personalization challenges in LLMs and introduces instruction-tuning
to improve model reasoning about reviewers' preferences.

**Strengths:**

The key strength of this research lies in its
investigation of the contamination issue in GPT-4 and its proactive
response by creating two new datasets, Per-MPST and Per-DOC. These
datasets are designed to mitigate contamination problems in story
evaluation, making this study a valuable contribution to personalized
text generation assessment.

**Weaknesses:**

The current text leaves room for confusion regarding the roles of
different models and lacks clarity in explaining the basis of metrics
for evaluation. Here are the three identified weaknesses:

Lack of Clarity in Model Attribution in contribution statements: The
paper needs to explicitly specify that "Current LLMs" refer to models
like ChatGPT-4, and it should distinguish that the pronoun "they" in
"with instruction-tuning on several thousands of data, they can"
pertains to the proposed method based on LLaMA-2. Failing to do so may
lead to ambiguity, making it difficult to understand the paper's
contributions.

Ambiguity in Metric Basis: While the paper mentions reporting
prediction accuracy as the primary metric for comparative evaluation,
it doesn't make it clear what this accuracy is based on. The reference
to "prediction accuracy" needs to be explicitly linked to the
mentioned Pearson correlation or other relevant metrics, providing a
clear understanding of what's being measured.

Limited Diversity in Baseline Models: The paper primarily relies on
LLM-based approaches for score prediction, neglecting the inclusion of
various other methods commonly used in the field, such as matrix
factorization and other score (rating) prediction techniques. To
provide a more comprehensive evaluation, it would be beneficial to
incorporate these alternative baseline models for a more thorough
comparison.


By addressing these issues, the paper can enhance its clarity and
comprehensiveness, making the contributions and methodology more
apparent to readers.

**Questions:**

Ambiguity in Metric Basis: The paper mentions "prediction
accuracy" as the primary metric for comparative evaluation. Could you
please clarify what specific metrics this "prediction accuracy" is
based on? Is it linked to the previously mentioned Pearson
correlation, or are there other relevant metrics involved? Providing
this clarification will enhance the understanding of the evaluation
process.

Limited Diversity in Baseline Models: The paper predominantly focuses
on LLM-based approaches for score prediction. Are there any specific
reasons for not including a wider range of commonly used methods in
the field, such as matrix factorization or other score prediction
techniques, as part of the baseline models for comparison? Including
these alternatives could potentially offer a more comprehensive
evaluation. Could you share your insights on this choice?


PERSE is a method fine-tuned on the LLAMA-2 using proposed
datasets. However, chatGPT-4, while being a formidable model, does not
undergo fine-tuning in the same way. Do you think it would be fairer
to apply fine-tuning to chatGPT-4 via OpenAI's API to ensure a level
playing field?

---

> ### Author Response · Authors · 2023-11-18
> **Thanks for the valuable comments! Here we discuss these concerns as below**
>
> **Q1.  Ambiguity in Metric Basis: clarify what specific metrics this "prediction accuracy" is based on. Is it linked to the previously mentioned Pearson correlation, or are there other relevant metrics involved?**
>
> A1. The prediction accuracy is the accuracy of binary classification, which is calculated by $accu = \frac{\text{the number of true prediction}}{\text{the total example size}}$. We view the comparison of each aspect as a binary classification between the given two plots. We also clarify this point in the revised manuscript.
>
> ---
>
> **Q2. Lack of clarity in model attribution in contribution statements. "Current LLMs" refer to models like ChatGPT-4, and it should distinguish that the pronoun "they" in "with instruction-tuning on several thousands of data, they can" pertains to the proposed method based on LLaMA-2.**
>
> A2. Here the “current LLMs” refer to the direct usage of LLMs as the evaluation model, and “they can” indicates the instruction-turned LLMs. Both of them can refer to LLaMA-based or GPT-4-based models. The main difference here is before and after instruction-turning. We have revised the contribution part.
>
> ---
>
> **Q3. Limited Diversity in Baseline Models: For not including a wider range of commonly used methods in the field, such as matrix factorization or other score prediction techniques, as part of the baseline models for comparison?**
>
> A2. We add matrix factorization baseline[1] on Per-MPST for the main Table 2 (previous Table 3) as below. However, on Per-Doc, both plot pairs and the annotators of the test set have no overlapping with the training set, so the matrix factorization cannot apply to this setting. MF baseline has a similar performance to the simple baseline, which averages the score in the preference.
>
> |             | Pearson | Spearman | Kendall |
> |-------------|---------|----------|---------|
> | simple baseline    |   0.301 |    0.302 |   0.230 |
> | MF          |   0.308 |    0.313 |   0.269 |
> | llama-2-7b  |   0.146 |    0.117 |   0.094 |
> | llama-2-13b |   0.172 |    0.182 |   0.147 |
> | llama-2-70b |   0.214 |    0.232 |   0.181 |
> | GPT-4       |   0.315 |    0.312 |   0.253 |
> | PerSE-7b    |   0.307 |    0.329 |   0.263 |
> | PerSE-13b   |   0.345 |    0.368 |   0.293 |
>
>
>
>
> ---
>
> **Q4. Do you think it would be fairer to apply fine-tuning to chatGPT-4 via OpenAI's API to ensure a level playing field?**
>
> A4. Yes, we agree it would be a fairer comparison, and we expect that using PerSE with a more capable backbone model than LLaMA (e.g., ChatGPT) will naturally yield better performance downstream because the method we proposed in PerSE is orthogonal to the backbone. We included ChatGPT not as necessarily a fair comparison but just as a point of reference to put PerSE's performance in context.
>
>
> [1] Matrix factorization techniques for recommender systems

---

### Official Review · Reviewer_jhUM · 2023-10-31

**Soundness:** 2 fair
**Presentation:** 3 good
**Contribution:** 3 good
**Rating:** 5
**Confidence:** 4

**Summary:**

In this paper, the authors present a study on the interesting problem of personalized evaluation of open-ended story generation. Concrete methods are proposed to create two personalized datasets of story evaluation from existing datasets such as IMDB plots, in order to resolve the problem of data contamination in Large Language Models (LLM) evaluation models. The story evaluation datasets are generated for comprehensive personalized evaluation with proper anonymization and new personalized labels. More importantly, a personalized story evaluation model based on LLM is developed to infer reviewer preferences and provide a personalized evaluation. Experiment results show the LLaMA instruction-tuning based implementation achieves a high correlation with the reviewer’s judgment, outperforming GPT-4 in terms of specific metrics such as Kendall correlation of story ratings and pairwise preference prediction accuracy.

**Strengths:**

1. This paper is well written with concrete examples. For example, many examples of premises, plots and preference evaluations are given to explain personalized evaluation of story generation.

2. The problem of personalized story evaluation is under-explored and hence the novelty of this paper is high. The study in this paper may inspire many further research works in this domain.

3. Comprehensive experiments are conducted to illustrate the effects of major factors in modeling personalized story evaluation. And the metrics to evaluate the models are diverse.

**Weaknesses:**

1. There are missing technical details such as how exactly an anonymized plot is summarized.

2. The datasets generated seem to be in a small scale. To show the proposed methods are scalable and widely applicable, it'd be great if much larger scale datasets could be used.

**Questions:**

1. What is the significance of the comparison results between PERSEcomp-13b and PERSEcomp-7b in Table 4? Providing t-test results can be useful.

---

> ### Author Response · Authors · 2023-11-18
> **Thanks for the valuable comments! Here we discuss these concerns as below**
>
> **Q1. how exactly an anonymous plot is summarized?**
>
> A1. We use oasst-30b[1] to summarize the plots by prompting it to give a summary of the main idea of the anonymous plot. We add a flowchart for data construction in Figure 2 and provide the detailed prompts we use for anonymization and summarization in Appendix Figure 9. We also refine the description in Section 3 to make it clearer.
>
> ---
>
> **Q2. It'd be great if much larger scale datasets could be used.**
>
> A2. One of our motivations for re-proposing Per-MPST and Per-DOC is the difficulty of collecting personalized evaluation data. Because of this difficulty, the fact that PerSE still works well with our current smaller dataset sizes is also a strength of our method - it doesn't need the larger datasets to work fine. Moreover, our Per-MPST has ~10k data for training and ~1k for testing, which is comparable to other personalized text classification and generation datasets[2].
>
> ---
>
> **Q3. What is the significance of the comparison results between PERSEcomp-13b and PERSEcomp-7b in Table 4? Providing t-test results can be useful.**
>
> A3. We perform the t-test between PERSEcomp-13b and PERSEcomp-7b for Table 4 with *scipy.stats.ttest_ind* and report the p-value here. All p values are smaller than 0.05, indicating that these two models significantly differ from each other.
>
> |    | p-value  |
> |----|----------|
> | Q1 | 2.78E-10 |
> | Q3 | 1.50E-06 |
> | Q4 | 3.32E-05 |
> | Q5 | 3.12E-08 |
> | Q6 | 8.65E-05 |
>
>
> ---
>
> [1] OpenAssistant Conversations--Democratizing Large Language Model Alignment.
> [2] LaMP: When Large Language Models Meet Personalization

---

### Official Review · Reviewer_9dNi · 2023-10-31

**Soundness:** 3 good
**Presentation:** 3 good
**Contribution:** 4 excellent
**Rating:** 8
**Confidence:** 5

**Summary:**

The paper proposes datasets and LLaMA-2 based models (PerSE) for evaluating story generation in a personalized manner. This is motivated by an analysis in GPT4 evaluations which shows a marked bias toward known plot lines. The paper presents two kinds of evaluation setups, one based on predicting the rating a reviewer would assign a generated text in independence from other generations. And a pairwise aspect based evaluation that considers pairs of generations and predicts which of the two would be selected by a reviewer for a specific aspect (interestingless, surprise etc). The proposed datasets leverage existing datasets of movie reviews and pairwise annotated aspects to train PerSE. The paper then presents an analysis of the proposed models.

**Strengths:**

- The paper fills an important gap in the evaluation of text generation systems - one that is likely to be important in the near future as personalized generation becomes more important.
- The experiments presented are comprehensive and largely convincing.

**Weaknesses:**

- The paper could benefit from a discussion of some factors about the practicalities of using an approach like PerSE for evaluation; for example, what happens if there isn't training data for PerSE in a domain? How biased is PerSE toward LLaMa generations? others asked in the questions below.
- The paper could benefit from clarifications regarding some of its analysis and experiments.

**Questions:**

- What is the rationale for expecting that movie review data of MPST is not a part of pretraining data for LLaMA? Is this likely to lead PerSE models to be biased similarly to GPT4 (albeit to a lesser extent due to personalization)?
- Consider adding a citation/link for the MPST dataset.
- I found the extended analysis of Sec 3.1 in A.1.2 to be significantly easier to understand than the writing of Sec 3.1. Please consider rewriting this section for clarity and to stand alone in the main body of the paper.
- Relatedly, model based evaluations have been found to contain self-biases (https://arxiv.org/abs/2212.10020) - consider discussing or experimentally demonstrating the extent to which PerSE is likely to prefer generations from a LLaMA based text generation model.
- Sec 5.1: How are the k examples from a reviewer's historical reviews used for training and evaluation selected? Is this a random set of k examples for the reviewer or is something else done?
- Do I understand correctly that the DOC dataset contains generations from an existing system? If so, please discuss the implications of training PerSE on generated stories and its ability to judge generations from other systems. In a future use of PerSE do you envision that researchers will train a personalized evaluation model on the ratings from a small set of annotators?
- Continuing from my previous question - it seems like the paper primarily proposes a method for rating prediction and posits that this may be used for personalized story evaluation. Please discuss how you envision this actually being used for text generation evaluation in much more depth. For example, what happens when there is no training data for training a model like PerSE in a specific domain? Why does it make sense to use a model pretrained on one set of users to evaluate text generated for a (hypothetical) other set of users? - it seems the most sensible for the text generation model to be trying to personalize text to the specific users on who the evaluation model is trained.
- It is not clear to me that the the bias present in GPT4 for known plots immediately motivates a personalized evaluation - please consider rethinking/rewriting this motivation. For example, is there a simple bias correction one could apply to ratings produced from GPT4 so that known plots receive smaller scores?

---

> ### Author Response · Authors · 2023-11-18
> **Thanks for the valuable comments! Here we discuss these concerns as below**
>
> **Q1. What happens if there isn't training data for PerSE in a domain?**
>
> A1. In principle, PerSE is not specifically trained to evaluate stories because the main capability PerSE gains from finetuning is to analyze the reviewers’ preferences from the prior reviews and use these preferences to evaluate new cases. The story generation is a typical generation task where personalized evaluation is important. If there is no personalized training data for evaluation, we can directly prompt our personalized fine-tuned PerSE (as what we did on GPT-4 and LLaMA baselines) to evaluate. If there is a small amount of annotated personalized data in this domain, then we can further fine-tune PerSE on this domain to improve the performance.
>
> ---
>
> **Q2. How biased is PerSE toward LLaMa generations?**
>
> A2. Based on the description in the official repo [1], the story pairs in PerDOC are generated by a LLaMA-based fine-tuned model (oasst-30b). To investigate the potential bias in the story generation model, we randomly select one of the paired plots and ask GPT-4 to rewrite it. We randomly chose 50 examples for each aspect in Per-DOC. After rewriting one of the paired plots, we asked PerSE-7b and PerSE-13b to compare them again. We calculate the percentage that PerSE mistakenly selects the LLaMA’s output (the original plot) as the better one, and the total percentage that PerSE prefers LLaMA’s output. The results are shown below. PerSE-7b is a little biased on the LLaMA-based stories, while PerSE-13b has no obvious bias. It indicates that with finetuning our PerSE is less biased towards LLaMA-based generation.
>
> |           | PerSE 7b |PerSE 13b |
> | --------- | --------------------- | --------------------- |
> | the percentage of mistakenly preferring LLaMA’s stories  | 54.74%            |   48.51%         |
> | the total percentage of selecting LLaMA’s stories | 56.05%                | 49.60%                |
>
> ---
>
> **Q3. What is the rationale for expecting that movie review data of MPST is not a part of pretraining data for LLaMA? Is this likely to lead PerSE models to be biased similarly to GPT4 (albeit to a lesser extent due to personalization)?**
>
> A3. As we further discussed in previous Appendix A.1.2, LLaMA models have a similar bias as the GPT-4 (We moved this figure to the main body and it is Figure 3 now). This is why we use anonymization and summarization to alleviate the influence of memorization in our data construction. Therefore, our test set is less affected by the known bias and is a better benchmark to evaluate LLM’s capability of evaluation. Moreover, it ensures that our PerSE is tuned on a less biased training set.
>
> ---
>
> **Q4. Citation / link for the MPST dataset**
>
> A4.We fixed this in the revision.
>
> ---
>
> **Q5. Rewrite Section 3**
>
> A5. We have rewritten this section in our revision and see the general response for more details.
>
> ---
>
> **Q6. Discuss or experimentally demonstrate the extent to which PerSE is likely to prefer generations from a LLaMA-based text generation model.**
>
> A6. See A2 above.
>
> ---
>
> **Q7. Sec 5.1: How are the k examples from a reviewer's historical reviews used for training and evaluation selected? Is this a random set of k examples for the reviewer or is something else done?**
>
> A7. As described in Section 3, we randomly sample k reviews. It is used to simulate the inference time where the new reviewers randomly provide several reviews as their preference.
>
> ---
>
> **Q8. Discuss the implications of training PerSE on generated stories and its ability to judge generations from other systems.**
>
> A8. Here we would like to highlight that PerSE can evaluate stories generated by any system and the tuning phase will not introduce extra bias to the system it was trained on. There are several reasons:
>
> 1. As we show in A2, although PerSE is tuned on the stories generated by the LLaMA-based model, it does not show obvious preference over the LLaMA’s generations.
> 2. In our training data, all stories are generated by the same system (DOC framework with oasst-30b). Some generated stories are preferred and some are rejected. It is less likely for PerSE to learn a bias towards this system (for example, always give a better score on this system).
> 3. Even if we have multiple systems in our training data, our PerSE will not learn the bias either. One reason is that during the finetuning there is no system label to indicate where the stories are from, so PerSE cannot get such information and learn the bias. The other reason is that in the personalized setting, for the same premise with the same story pair, the preferred one could change due to the different reviewers’ preferences. So it is less likely for PerSE to learn a clear bias of always preferring one system.
> 4. Training on human ratings of the generated system outcome is a commonly used method in learned evaluation metrics and has shown great success and generalization ability in many metrics, such as BLEURT[2] and COMET[3].

---

> ### Author Response · Authors · 2023-11-18
>
> **Q9. In a future use of PerSE do you envision that researchers will train a personalized evaluation model on the ratings from a small set of annotators?**
>
> A9. As we clarify in General Response,  the small set of annotators and their preferences are organized as the instructions for instruction tuning [4]. The success of instruction-tuned LLMs (such as RLHF in GPT-4) shows that with a little data, LLMs can improve the capability of following the instructions. So in future use, researchers can use the training strategy of PerSE in other domains and improve the capability of evaluating from a personalized view.
>
> ---
>
> **Q10. Why does it make sense to use a model pre-trained on one set of users to evaluate text generated for a (hypothetical) other set of users?**
>
> A10. Two settings (predicting for existing reviewers / predicting for unseen reviewers) have their own use cases. For the first setting, the model already remembers the preferences of these reviewers and directly predicts for them. Under the second setting, the model can be easily adapted to new reviewers without further training. Our PerSE focuses on the second setting. It does not try to remember the preferences of these annotators. Instead, it predicts the preferences of new reviewers that are unseen during training. **In our test set, all reviewers are different from those in the training set.** The experimental results show that our PerSE can be used for new annotators with new preferences.
>
> ---
>
> **Q11. The bias present in GPT4 for known plots immediately motivates a personalized evaluation - please consider rethinking/rewriting this motivation. For example, is there a simple bias correction one could apply to ratings produced from GPT4 so that known plots receive smaller scores?**
>
> A11. To create a dataset for personalized story evaluation, we need to alleviate the influence of memorization to ensure the performance of different evaluation models is not affected by the known bias. We have revised this section to make it clearer and we refer you to the general response and the revision manuscript for detailed information. We propose two techniques: anonymization and summarization to reduce the known bias and use them during our dataset construction. Figure 3 shows the effectiveness of our strategies.
>
> ---
>
> [1] https://github.com/facebookresearch/doc-storygen-v2.
> [2] BLEURT: Learning Robust Metrics for Text Generation.
> [3] COMET: A Neural Framework for MT Evaluation.
> [4] The Flan Collection: Designing Data and Methods for Effective Instruction Tuning.

---

### Author Response · Authors · 2023-11-18
**General Response**

Thanks to all reviewers for their valuable comments!

We apologize for the confusion caused by our writing. We want to reiterate our goal:

*Our goal is to propose a method to evaluate story plots from the personalized perspective of a new reviewer, given only a few of this reviewer’s review examples*.

To address this problem, we introduce a dataset that is properly constructed so that its evaluation scores are less biased, and thus we need to mitigate the evaluation bias provided in GPT4. Our previous version can be confusing since we do not provide the context as to why we would like to reduce the evaluation bias in Sec. 3.1. We now fixed it in the new revision.

**The main modifications are listed below and highlighted in red in our revision. Note that the ids of Figure and Table change because of the reorganization.**

1. Modify claims in the introduction to reduce  Ambiguity (Thanks reviewer *7x1E*).
2. Rewrite Section 3 Personalized Story Evaluation Dataset, especially the bias section. Specifically, we remove the original bias analysis from Appendix A.1.2 and elaborate our dataset construction to make the whole pipeline clear. We also add a flowchart for the dataset in Figure 2 (Thanks reviewer *hjpK*). We moved Figure 3 from the Appendix to the main body because it is easier to understand (Thanks reviewer *9dNi*).
3. Add more baselines such as matrix factorization (Thanks reviewer *7x1E*) and t-test (Thanks *jhUM*).
4. Add a figure for our PerSE framework in Appendix Figure 11 (Thanks reviewer *hjpK*).

---

And here are two common concerns:

**Q1. The goal of PerSE’s finetuning and how to adapt PerSE to new reviewers?**.
A1. The goal of our PerSE is to predict personalized reviews for new reviewers that are never seen during the training phase. So it can be easily used for new reviewers without being re-trained on the preferences of these reviewers.
So during the instruction tuning phase, PerSE is tuned to follow the instruction to understand the reviewer’s preference from the k given examples (k is just 1 to 5) and use the preference to predict a personalized review for the new story. PerSE can be directly used for new reviewers, which is the setting of our testing: all reviewers in the test set are never seen during the training phase.

**Q2. The relation between contamination / memorization / bias and personalization**.
A2. To assess the performance of different personalized evaluation models, we need clear personalized story evaluation datasets that are less affected by the existing biases. The contamination section discusses several biases caused by the memorization issue which we would like to alleviate in the dataset construction. We revise the Section 3 to make it clearer.

---

> ### Author Response · Authors · 2023-11-21
> **Rebuttal Reminder**
>
> Thanks again for the insightful feedback on our work!
>
> We've carefully worked to address your comments/questions and revised our manuscript based on the valuable suggestions.
>
> We would like to kindly remind you that the end of the discussion phase is coming in two days and We are happy to discuss any further concerns.

---

### Author Response · Authors · 2023-11-22
**Summary of the concerns and response**

Dear AC and reviewers,

Thank you for handling our paper! We read all of the reviewers’ comments carefully, and have reorganized our paper and added new experiments and findings in the revised paper that are highlighted in red.

Here we would like to briefly summarize the reviewers’ concerns and our revision and are looking forward to more discussion.

We are glad to see that reviewers find our work on personalized story evaluation **novel and important (9dNi, jhUM, 7x1E)** and **experiments comprehensive and adequate (9dNi, jhUM, hjpK)**.

Based on the insightful comments, we further improve our manuscripts from the following aspects:

(1) As suggested by reviewers (*9dNi, 7x1E, hjpk*), we re-write Section 1 (introduction) and Section 3 (Personalization Story Evaluation Dataset) to reduce ambiguity. We also add a flowchart to demonstrate the pipeline of our data construction. More detailed revisions are shown below.

(2) Reviewer *9dNi* raised a concern that our LLaMA-based PerSE might have been biased towards LLaMA-based generation. We added experiments about the PerSE’s preference among LLaMA-based and GPT-4-based story generations and found there is no obvious bias after the instruction finetuning.

(3) Reviewer *9dNi and hjpK* commented on the generalization and future application of our PerSE. We highlighted that our model is tested on unseen users and the excellent results indicate that it can generalize well to new users. PerSE obtains the capability of following the formatted instructions to analyze personalized preferences and use them to evaluate. PerSE can be further used with RLHF to tailor the story generation for specific preferences and needs of different reviewers and can also help recommendation systems to suggest more suitable stories.

(4) Reviewer *7x1E* suggested adding more baselines and we added more traditional baselines in recommendation systems such as the matrix factorization.

(5) Reviewer *hjpK* suggested providing more illustrations to make the whole system clearer and we added a flowchart for dataset construction and a framework overview for PerSE.

Thanks again for your time and efforts! We would be grateful if you would take these into consideration when making the decision.
We would also appreciate it if reviewers could engage more with us on our responses to see whether their concerns have been solved.

---

### Meta-Review · Area_Chair_rpk3 · 2023-12-08

**Metareview:**

The research presents PerSE, a model designed for personalized story evaluation. It harnesses the unique preferences of reviewers to produce customized reviews for various stories. For model evaluation, the study introduces two datasets, on which PerSE outperforms GPT-4.

After the rebuttal period, two concerns still remain unresolved.

Firstly, the submission suffers from a lack of clarity, leading to questions about the dataset construction and evaluation process. Although the author's responses addressed some of the reviewers' questions, these clarity questions are essentially the consequence of the confusion during the review process. In my extensive experience with paper reviews, the rebuttal period is not typically intended to resolve writing issues. Therefore, it is clear that the paper requires careful revision to meet the publication standards.

Secondly, based on what we can understand for the current submission status, the focus of this submission is not entirely clear. (This might explain the diversity of the assigned reviewers). The author claims contributions in both the dataset and the proposed evaluation model, but both contributions appear to be insufficient:

- Dataset contribution: The potential applications of the proposed task have not been well explored and discussed. The task defined by the Per-MPST dataset does not introduce significant novelty to the field of personalized recommendation, which has been extensively studied for decades.
- Model contribution: Unlike standard prediction models, the proposed PerSE appears to function more as a metric rather than an end-task model. Therefore, it should be either evaluated in a transfer setting or, perhaps, considered as a reward model for tasks of story generation.

**Justification For Why Not Higher Score:**

Lack of clarity and vague contributions as discussed in the meta-review.

**Justification For Why Not Lower Score:**

NA

---

### Decision · Program_Chairs · 2024-01-16

Reject